# Structural and Kinetic Characterization of the SpeG Spermidine/Spermine *N*-acetyltransferase from Methicillin-Resistant *Staphylococcus aureus* USA300

**DOI:** 10.3390/cells12141829

**Published:** 2023-07-12

**Authors:** Sofiya Tsimbalyuk, Aleksander Shornikov, Parul Srivastava, Van Thi Bich Le, Imani Warren, Yogesh B. Khandokar, Misty L. Kuhn, Jade K. Forwood

**Affiliations:** 1School of Dentistry and Medical Sciences, Charles Sturt University, Boorooma Street, Wagga Wagga, NSW 2678, Australia; stsimbalyuk@csu.edu.au (S.T.); parulsahaj19@gmail.com (P.S.); khandoky@ansto.gov.au (Y.B.K.); 2Deparment of Chemistry and Biochemistry, San Francisco State University, San Francisco, CA 94132, USA; 3School of Biomedical Sciences, Charles Sturt University, Wagga Wagga, NSW 2678, Australia

**Keywords:** Gcn5-related *N*-acetyltransferase (GNAT), *Staphylococcus aureus*, acetylation, spermidine/spermine N-acetyltransferase (SSAT), polyamine, MRSA USA300

## Abstract

Polyamines are simple yet critical molecules with diverse roles in numerous pathogenic and non-pathogenic organisms. Regulating polyamine concentrations affects the transcription and translation of genes and proteins important for cell growth, stress, and toxicity. One way polyamine concentrations are maintained within the cell is via spermidine/spermine *N*-acetyltransferases (SSATs) that acetylate intracellular polyamines so they can be exported. The bacterial SpeG enzyme is an SSAT that exhibits a unique dodecameric structure and allosteric site compared to other SSATs that have been previously characterized. While its overall 3D structure is conserved, its presence and role in different bacterial pathogens are inconsistent. For example, not all bacteria have *speG* encoded in their genomes; in some bacteria, the *speG* gene is present but has become silenced, and in other bacteria, it has been acquired on mobile genetic elements. The latter is the case for methicillin-resistant *Staphylococcus aureus* (MRSA) USA300, where it appears to aid pathogenesis. To gain a greater understanding of the structure/function relationship of SpeG from the MRSA USA300 strain (SaSpeG), we determined its X-ray crystal structure in the presence and absence of spermine. Additionally, we showed the oligomeric state of SaSpeG is dynamic, and its homogeneity is affected by polyamines and AcCoA. Enzyme kinetic assays showed that pre-incubation with polyamines significantly affected the positive cooperativity toward spermine and spermidine and the catalytic efficiency of the enzyme. Furthermore, we showed bacterial SpeG enzymes do not have equivalent capabilities to acetylate aminopropyl versus aminbutyl ends of spermidine. Overall, this study provides new insight that will assist in understanding the SpeG enzyme and its role in pathogenic and non-pathogenic bacteria at a molecular level.

## 1. Introduction

Antibiotic resistance is an urgent global health threat causing an estimated ~5 million deaths worldwide in 2019, and it is crippling our ability to treat bacterial infections effectively [1]. One of the culprits in this struggle includes methicillin-resistant *Staphylococcus aureus* (MRSA) bacterial infections, which can be either community associated (CA) or hospital associated (HA). In the late 1990s, the CA-MRSA USA300 strain spread rapidly through the US and became the leading cause of skin and soft tissue infections. Its emergence led not only to the displacement of other *S. aureus* strains but also to an overall increase in the incidence of MRSA infections [2]. Methicillin is a β-lactam antibiotic, and its resistance in *S. aureus* is conferred by a penicillin-binding protein 2a that is encoded by *mecA* and carried on the low fitness cost staphylococcal chromosomal cassette *mec* (SCC*mec*) [3,4,5].

The USA300 MRSA strain is unique in its reduced sensitivity to polyamines, which has been thought to contribute to its enhanced fitness [6,7]. Polyamines are small, positively charged molecules that play important roles in tissue repair [8]. Anti-inflammatory responses include an increase in host polyamine production, which aids in clearing *S. aureus* in skin infections, but has a notably lower effect on USA300 strains. It has been assumed that polyamines are universally distributed and necessary for cellular functions in all domains of life. However, bacteria do not exhibit a conserved role for polyamines, and many lack the biosynthetic genes for their synthesis but retain import and export mechanisms [9,10]. Indeed, *Staphylococci* are capable of importing and modifying various polyamines [9,11], but polyamine biosynthetic genes are absent in the *S. aureus* USA300 strain, and it does not produce endogenous spermine, spermidine, or homospermidine [6,9].

Intriguingly, Rozansky et al. observed polyamine toxicity toward *S. aureus* [12], but the USA300 strain is able to overcome this toxicity by regulating intracellular concentrations of spermine and spermidine. This ability arose via the acquisition of an arginine catabolic mobile element (ACME) from a successful skin colonizer and opportunistic pathogen *Staphylococcus epidermidis,* through horizontal gene transfer [13]. While there is significant diversity in organization and presence of genes within ACME in *S. epidermidis*, the similarity of ACME genes found in the *S. aureus* USA300 strain indicates a single transfer event at the end of the twentieth century, which was estimated to occur between 1981 and 1997 [7]. One of the genes on ACME thought to contribute to reduced polyamine toxicity in the USA300 strain is *speG* which encodes a spermidine/spermine acetyltransferase (SSAT) SpeG. SSATs in a variety of organisms have critical roles in regulating polyamine intracellular concentrations, and Li et al. recently showed that SpeG from *S. aureus* acetylates diverse polyamines [9].

The presence of *speG* in MRSA strains varies widely across ACME type I, type II, and type II’ elements, and other SCCs, or it can be completely absent [14,15]. Diep et al. 2008 previously observed that when ACME is deleted, the fitness of *S. aureus* USA300 is significantly reduced [3]. While initial results with the USA300 strain pointed toward a connection between bacterial survival and the presence of *speG*, new studies have questioned this perspective. Debate ensues regarding whether or not *speG* is required for virulence and its role or lack thereof in pathogenesis. Montgomery et al. 2009 showed virulence is not affected when *speG* was deleted [16], but Thurlow et al. 2013 showed it was required for virulence in both the Los Angeles Clone [LAC] and San Francisco 8300 [SF8300] clones [8]. Later, Wu et al. showed ACME alone does not support *S. aureus* USA300 bacterial virulence [17]. Overall, it appears that the key to the USA300 strain being less sensitive to polyamines is the presence of *speG*, but it is not known if its enzymatic activity or some other property of this protein is the critical factor for its retention within various MRSA strains. During all growth stages of *S. aureus* USA300, SpeG is constitutively expressed, regardless of exogenous spermine concentrations [6]. Thus, it appears that SpeG is not produced in response to polyamines, which is puzzling since it is thought that *speG* was acquired to acetylate polyamines to completely negate their bactericidal effects. Since the acquisition of *speG* by a variety of pathogenic bacteria has implicated it in critical aspects of antibiotic resistance, pathogenesis, and bacterial survival, we sought to study the structure/function relationship of SpeG from the *S. aureus* USA300 strain (SaSpeG) to gain greater insight into this enzyme from this globally problematic pathogen.

## 2. Materials and Methods

### 2.1. SpeG Expression and Purification for Protein Crystallization and Initial Substrate Screening Assay

The recombinant *Staphylococcus aureus speG* gene was synthesized commercially by Genscript and cloned into the pMCSG21 expression vector at the SspI site (UniProt ID: A0A0H2XGJ0). The recombinant plasmid was transformed into *E. coli* BL21 (DE3) pLysS cells and a single colony were used to inoculate 5 mL of the Luria Bertani (LB) broth containing spectinomycin at a concentration of 100 µg/mL. This culture was grown overnight in a shaking incubator at 37 °C and used to inoculate 2 L of autoinduction media (5% tryptone, 10% yeast extract, Edwards Group Pty Ltd., Narellan, Australia) containing spectinomycin at a concentration of 100 µg/mL [18]. Cells were grown at 25 °C with shaking at 120 rpm for 24 h, then harvested by centrifugation at 18 °C, 6000× *g* rpm for 30 min. The cell pellet was resuspended in a binding buffer containing 100 mM sodium phosphate pH 8.0, 300 mM NaCl, and 20 mM imidazole and stored at −20 °C. Cells were lysed by three freeze–thaw cycles in the presence of 1 mL of lysozyme 20 mg/mL, and cell debris was removed by centrifugation at 18 °C, 12,000× *g* rpm for 30 min. The supernatant was passed through a 0.45 µm syringe filter (Millipore Sigma, Burlington, MA, USA) before loading onto a 5 mL HisTrap nickel-affinity column (GE Healthcare, Marlborough, MA, USA). The column was washed with ten CVs of binding buffer and eluted with a 100% gradient over five column volumes with buffer containing 100 mM sodium phosphate pH 8, 300 mM NaCl, and 500 mM imidazole. The eluted protein was incubated with 100 µL TEV protease (5 mg/mL) for 12 h to remove the His-tag, then purified by size exclusion chromatography using a pre-equilibrated 26/60 S200 column (GE Healthcare, Marlborough, MA, USA) with 50 mM Tris pH 8.0 and 150 mM NaCl. A total of ~13 mg of protein was obtained from 4 L of bacterial culture, which was then concentrated to a final concentration of ~10 mg/mL and stored at −80 °C. The purity, assessed by SDS-PAGE and size exclusion chromatography, was >95%.

### 2.2. Crystallization and Structure Solution

SaSpeG was screened both in ligand-free form and in complex with spermine using the hanging drop vapor diffusion method and the commercially available Hampton Research Screens, Crystal Screen (HCS-1), Crystal Screen 2 (HCS-2), PEG/Ion, and PEG/Ion 2. Three times the molar concentration of spermine (Sigma-Aldrich, an affiliate of Merck KGaA, Darmstat, Germany) was used for screening. Initial crystals were observed in conditions 4 and 37 of HCS-1 and 23, 25, 26, and 32 of HCS-2. These conditions were optimized by varying pH, precipitant concentration, and protein concentration. The crystallization condition that led to structure determination of apo SaSpeG and SaSpeG in complex with spermine was 0.1 M NaCl, 0.1 M HEPES pH 8.0, and 1.6 M ammonium sulfate. A single crystal was cryoprotected in mother liquor containing 20% glycerol, and X-ray diffraction data were collected at the Australian Synchrotron MX1 and MX2 beamlines. The data were indexed and integrated using Mosflm [19] and scaled in aimless [20]. Molecular replacement was performed using chain A of model PDB ID 4JJX in Phaser [21]. Structural refinement and model building were performed using Refmac [22], Phenix [23,24], and Coot [25].

### 2.3. Protein Expression and Purification for SaSpeG Steady-State Kinetic Characterization

The spectinomycin-resistant plasmid containing the *S. aureus speG* gene was transformed into *E. coli* BL21 (DE3) cells and a single colony were selected to make a glycerol stock. This stock was used to inoculate a 5 mL starter culture of LB, which was grown overnight at 37 °C with shaking. A volume of 200 mL of Terrific Broth in a 1 L shaker flask was then inoculated with 2 mL of starter culture and grown until the OD_600nm_ reached 0.6. Afterward, the culture was cooled on ice, and 0.5 mM IPTG was added to induce protein expression overnight at RT with shaking. The concentration of spectinomycin throughout all growth steps was 50 µg/mL. Cells were harvested by centrifugation and resuspended in 30 mL of lysis buffer (10 mM Tris-HCl pH 8.3, 500 mM NaCl, 5 mM imidazole, 5% glycerol, and 5 mM beta-mercaptoethanol (BME)), sonicated, and stored at −80 °C as described previously [26]. Crude extracts were thawed and centrifuged at 25,000× *g* for 45 min at 4 °C. The supernatant was loaded onto a 1 mL HiTrap FF nickel affinity column (GE Healthcare) that was pre-equilibrated with 10 mM Tris-HCl pH 8.3, 500 mM NaCl, and 5 mM BME using an AKTA Start FPLC. The column was washed with buffer containing 10 mM Tris-HCl pH 8.3, 500 mM NaCl, 5 mM BME, and 25 mM imidazole, and the protein was eluted with buffer containing 10 mM Tris-HCl pH 8.3, 500 mM NaCl, 5 mM BME, and 500 mM imidazole. Afterward, the imidazole was removed using a PD-10 gravity flow column (GE Healthcare, Marlborough, MA, USA), and the protein was eluted into a buffer containing 10 mM Tris-HCl pH 8.3, 500 mM NaCl, and 5 mM BME and concentrated in an Amicon centrifugal device with a 10 kDa MWCO to 19.4 mg/mL. The molecular weight of the SaSpeG monomer was ~19.8 kDa, and the extinction coefficient of 26,360 M^−1^cm^−1^ was used to calculate the protein concentration.

Since we previously showed the activity of SpeG from *Vibrio cholerae* (VcSpeG) was decreased in the presence of the polyhistidine tag [27], we removed the tag on SaSpeG to perform kinetic assays. This was achieved by incubating the SaSpeG protein in a 20:1 ratio with TEV protease that was purified under similar conditions. The two proteins were combined and placed in a Slide-a-Lyzer mini-dialysis device (Thermo Fisher Scientific, Waltham, MA, USA) with cleavage buffer (50 mM Tris-HCl pH 8.3, 300 mM NaCl, 5 mM BME, 3.5 mM DTT, and 5% glycerol) and incubated at 37 °C for 2 h on an oscillating shaker. The cleavage buffer was exchanged, and the protein was moved to 4 °C to continue cleaving overnight. The protein was centrifuged to remove precipitate, and the supernatant was diluted with buffer (10 mM Tris-HCl pH 8.3, 500 mM NaCl, and 5 mM BME) in a 1:3 ratio of protein to buffer to reduce the concentration of DTT. Protein was then loaded onto the nickel affinity column using the procedure as above, except the protein was eluted using a gradient of buffer containing imidazole from 0–150 mM. The cleaved protein was eluted in two fractions: (1) flow-through after loading the cleaved protein onto the column and (2) eluate during the imidazole gradient. The tag cleavage was confirmed for both fractions by SDS-PAGE, but the amount of protein from the imidazole gradient was greater and therefore was exchanged into buffer composed of 10 mM Tris-HCl pH 8.3 and 500 mM NaCl and concentrated as described above to 3.8 mg/mL for enzyme kinetics assays.

### 2.4. SaSpeG Kinetic Characterization

The SaSpeG enzyme was initially screened for enzyme activity toward a modified broad panel of eighty small molecules to identify ligands for co-crystallization by the Forwood laboratory. Enzyme activity was measured spectrophotometrically using a discontinuous colorimetric reaction with 5,5′-dithiobis(2-nitrobenzoic acid) (DTNB) [27]. Each reaction was performed in a volume of 50 μL with 50 mM Tris-HCl pH 8.0, 0.5 mM AcCoA, and 25 mM substrate with the exception of acetyl-Ser-Asp-Lys-Pro, AICAR, and poly-L-lysine substrates, as their final concentrations in the reactions were 0.4 mM, 1.9 mM, and 4 mg/mL, respectively. The reaction was started by the addition of 10 μL (1 μg) of the enzyme, incubated for 10 min at 37 °C, and quenched by adding 50 μL of a solution composed of 100 mM Tris-HCl pH 8.0 and 6 M guanidine HCl. The enzymatic activity was recorded as the increase in the formation of the 2-nitro-5-thiobenzoate anion (TNB^2−^) at A_415nm_ (E = 13,600 M^−1^cm^−1^) by the reaction of DTNB with free CoASH. Later a collaboration was initiated where the Kuhn laboratory re-screened a small, tailored collection of compounds based on the results of the broad screening assay from the Forwood laboratory (agmatine, cadaverine, putrescine, spermine, and spermidine) to identify substrates that should be used for further kinetic characterization. For this screening assay, each 50 μL reaction contained 70 mM Bicine pH 8.0, 20 mM NaCl, 3 mM polyamine, 0.5 mM AcCoA, and 10 μL (0.35 μg) of enzyme and was allowed to proceed for 10 min at 37 °C. Similar to the previous work with the *Vibrio cholerae* SpeG enzyme [28], only spermine and spermidine were viable substrates at lower concentrations of the enzyme. Therefore, only these two substrates were used for further kinetic characterization of the SaSpeG enzyme with the two assays described below. All assays were performed in triplicate and the average of two technical replicates were reported.

#### 2.4.1. Assay 1: (Standard Assay) No Pre-Incubation of Enzyme with Polyamine

Substrate saturation curves of SaSpeG toward spermidine and spermine were performed as follows. Each 50 μL reaction contained 70 mM Bicine pH 8.0, 20 mM NaCl, 0.5 mM AcCoA, and polyamine (varied from 0–10 mM). A total of 10 μL of cleaved SaSpeG enzyme (0.14 μg) was used to initiate reactions. The reactions were performed at 37 °C for 5 min and terminated with 50 μL of solution (100 mM Tris-HCl pH 8.0 and 6 M guanidine HCl). 200 μL of a solution containing 100 mM Tris-HCl pH 8.0, 1 mM EDTA, and 0.2 mM DTNB was added, allowed to react for 10 min at RT, and then the CoA product was determined colorimetrically (A_415nm_), and data were fitted as described previously [26]. L-cysteine instead of CoA was used for standards.

#### 2.4.2. Assay 2: Pre-Incubation of Enzyme with Polyamine

Substrate saturation curves of SaSpeG toward spermine, spermidine, and AcCoA were performed by preincubating the cleaved enzyme with polyamine using a similar approach as described before [29] with the following modifications. Prior to performing enzyme kinetics assays, the purified and concentrated SaSpeG enzyme was diluted to 100 μg/mL in a solution containing 70 mM Bicine pH 8.0, 40 mM NaCl, 0.01% Triton-X 100, and 4 mM spermine or spermidine and stored in aliquots at −80 °C. These enzyme stocks were then used for subsequent steps of the experiment. Each 100 µL reaction contained 70 mM Bicine pH 8.0, 20 mM NaCl, and 0.005% Triton X-100 and was allowed to proceed for 5 min at 22 °C. When performing substrate saturation curves for polyamines spermidine and spermine (0–1.5 mM), the concentration of AcCoA was held constant at 1 mM. Alternatively, when substrate saturation curves for AcCoA (0–1.5 mM) were performed, the polyamine concentration of spermine or spermidine was held constant at 1.5 mM. Reactions for polyamine and AcCoA substrate saturation curves were initiated with AcCoA. The final amount of enzyme in each reaction was 2 ng for assays with spermine and 5 ng for assays with spermidine. While Assay 2 is reminiscent of the assay in Tsimbalyuk et al. [29], we further altered this assay to replace bovine serum albumin (BSA) with Triton X-100. This is because BSA is known to become acetylated, and we wanted to reduce the possibility of non-enzymatic acetylation of BSA with AcCoA during our reactions.

#### 2.4.3. Polyamine Substrate Saturation Curves via Assay 2

The SaSpeG enzyme stored with polyamine was diluted with enzyme dilution buffer (70 mM Bicine pH 8.0, 40 mM NaCl, 0.01% Triton X-100, and 0.4 mM polyamine) to prepare a 40X enzyme stock solution. This solution was then mixed with varying concentrations of polyamine substrates to give a 2X enzyme-polyamine solution (70 mM Bicine pH 8.0, 40 mM NaCl, 0.01% Triton X-100, 2X enzyme) at each polyamine concentration being tested. 50 μL of each 2X enzyme-polyamine solution was pipetted into individual wells of a 96-well polystyrene microplate, and the reaction was initiated with 50 μL of a 2X AcCoA solution (2 mM AcCoA, 70 mM Bicine pH 8.0). Reactions were terminated by adding 100 µL of stop solution (100 mM Tris-HCl pH 8.0 and 6 M guanidine HCl), and the amount of product was determined by an absorbance reading at A_405nm_ following the addition of 100 µL of a solution containing 0.4 mM DTNB in 100 mM Tris HCl pH 8.0 and 1mM EDTA.

#### 2.4.4. AcCoA Substrate Saturation Curves via Assay 2

The stock SaSpeG enzyme stored in the presence of polyamine was diluted to make a 2X enzyme-polyamine solution (70 mM Bicine pH 8.0, 40 mM NaCl, 0.01% Triton X-100, 1.5 mM polyamine, 2X enzyme). A series of 2X AcCoA solutions at varying concentrations of AcCoA (0–3 mM AcCoA in 70 mM Bicine pH 8.0) were also prepared. Similar to assays for generating polyamine substrate saturation curves, 50 μL of each solution (2X AcCoA solution and 2X enzyme-polyamine solution) were combined to initiate the reactions and were processed as described above.

### 2.5. SpeG Spermidine Substrate Specificity LCMS Assays

The specificity of the several SpeG enzymes (SaSpeG from *Staphylococcus aureus,* BtSpeG from *Bacilus thuringiensis,* EcSpeG from *Escherichia coli,* and VcSpeG from *Vibrio cholerae*) for N^1^ vs. N^8^ amines of spermidine was assessed using the following procedure. All proteins were expressed and purified as described previously and above [26,29,30]. The cleaved SpeG enzymes (9 ng of SaSpeG, 15 ng of BtSpeG, 25 ng of EcSpeG, and 25 ng of VcSpeG) were reacted with 2 mM spermidine and 1 mM AcCoA in 70 mM Bicine pH 8.0, 20 mM NaCl, and 0.005% Triton X-100 in a total reaction volume of 100 μL for 5 min at 22 °C. The reactions were stopped with 25 μL of 1 M guanidine HCl, and then reaction products were analyzed as described previously [29]. Briefly, polyamines were derivatized using dansyl chloride and were analyzed using a 1290 UPLC system equipped with a 6130 single quadrupole mass spectrometer (Agilent Technologies, Santa Clara, CA, USA). Standards included N^1^ and N^8^-acetylspermidine dihydrochloride (Millipore Sigma Cat# 01467 and Santa Cruz Biotechnology, Dallas, TX, USA Cat# SC-236151A, respectively) and spermidine trihydrochloride (Millipore Sigma, Burlington, MA, USA, Cat#85578). Data reported were consistent across two technical replicates.

### 2.6. Native PAGE Assays

Recombinant SaSpeG protein for Native PAGE was purified and cleaved as described for steady-state kinetic characterization (above) and then diluted with buffer containing 100 mM Bicine pH 8.0 and 100 mM NaCl. Reactions in the presence and absence of substrates were prepared by incubating varying concentrations of diluted SaSpeG and different concentrations of spermine and/or AcCoA in a total volume of 10 µL for 5 min. Concentrations of SaSpeG protein ranged from 1.25–25 µM, and AcCoA and spermine concentrations ranged from 0–2.5 mM. An equivalent volume of BioRad 2X Native sample buffer (62.5 mM Tris-HCl pH 6.8, 40% glycerol, 0.01% bromophenol blue) was then added to each reaction and mixed. Half of the sample (10 µL) was loaded onto a BioRad Any kD TGX Stain-Free gel and run at 200 V for 50 min in 1X BioRad running buffer (25 mM Tris, 192 mM glycine, pH 8.3). The gel was then stained with GelCode Blue Stain Reagent (Thermo Fisher Scientific) and destained with water.

### 2.7. Electrophoretic Mobility Shift Assays (EMSAs)

Recombinant SaSpeG protein for EMSAs was purified and cleaved as described for crystallization (above). Samples (10 µL) containing SaSpeG (0.2 mM) and/or polyamine (spermine or spermidine; 2 mM) and/or AcCoA (2 mM) and 3 µL of 50% glycerol were combined, and then half of the sample was loaded onto a 1.5% agarose gel in TB buffer (45 mM boric acid, 45 mM Tris base pH 8) and run for 12 h at 40 V. The gel was then stained by Coomassie brilliant blue and imaged using a Gel Doc XR+ system.

### 2.8. Analytical Size-Exclusion Chromatography (SEC)

SaSpeG protein produced for crystallization trials was used to perform SEC with a Superdex 200 10/300 GL analytical column (GE Healthcare) in buffer (50 mM Tris pH 8.0, 125 mM NaCl) in the presence and absence of different substrates. Samples (1 mL) contained 37.5 µM SaSpeG enzyme and/or 0.75 mM spermine or spermidine and/or 0.4 mM AcCoA in GST buffer A (50 mM Tris pH 8.0, 125 mM NaCl) and 500 μL was loaded onto the column. Aliquots of samples prior to loading and different fractions were collected and analyzed by SDS-PAGE. A total of 20 µL of protein from each fraction was mixed with an equivalent volume of SDS dye, and then 10 µL of the sample was loaded onto 4–12% Bis-Tris gel. Gels were run at 165 V for 30 min and stained with Coomassie blue.

## 3. Results

### 3.1. SaSpeG Kinetic Characterization

Since the SaSpeG protein was predicted to be responsible for protecting the MRSA USA300 strain from the bactericidal effects of polyamines, we performed an initial broad-substrate screen at high concentrations of substrates to determine which molecules should be further screened under steady-state conditions (Appendix A). From this screening assay, we identified activity toward spermine, spermidine, cadaverine, putrescine, agmatine, and N-acetyl spermine (Appendix A). While the initial broad screening assay showed activity toward high concentrations of agmatine, putrescine, and cadaverine, we did not observe activity toward these compounds below 3 mM or at lower concentrations of the enzyme under steady-state conditions. Therefore, we focused our characterization on spermine and spermidine acetylation using our standard enzyme assay (Assay 1) and found the pH optimum of the SaSpeG enzyme was pH 8.0.

Our previous work showed that the Gram-positive *Bacillus thuringenesis* (BtSpeG) enzyme exhibited a higher apparent affinity for polyamines and turnover when the enzyme was pre-incubated with polyamine [29]. Therefore, we used two assays to compare the effects of pre-incubation (Assay 2) and no pre-incubation (Assay 1) with polyamines on enzyme activity and kinetic parameters for the SaSpeG enzyme. Assay 1 showed the apparent affinity of the SaSpeG enzyme for both spermine and spermidine was in the millimolar range, and cooperativity increased when SaSpeG was not pre-incubated with a polyamine (Figure 1A and Table 1). The enzyme exhibited a 3-fold higher apparent affinity (1.3 mM vs. 4 mM for spermine and spermidine, respectively) and ~5-fold higher catalytic efficiency for spermine than spermidine. On the other hand, when we pre-incubated the SaSpeG enzyme with the respective polyamine prior to performing kinetic assays (Assay 2), we observed a dramatic increase (nearly 2-orders of magnitude) in apparent affinity for both spermine and spermidine and ~3-orders of magnitude increase in catalytic efficiency toward both spermine and spermidine compared to no pre-incubation (Assay 1) (Figure 1B,C and Table 1). Furthermore, Assay 2 results showed the apparent affinity of SaSpeG for AcCoA in the presence of saturating concentrations of spermine or spermidine was also comparable (Figure 1D). Thus, pre-incubation with polyamine significantly alters the kinetic behavior of the SaSpeG enzyme and appears to trigger a more “primed and ready” state for efficient acetylation of polyamines.

With the exception of spermidine substrate saturation curves, all other curves obtained using Assay 2 (pre-incubation with polyamine) showed Michaelis–Menten behavior with no cooperativity. However, while the positive cooperativity decreased for the spermidine substrate saturation curve in Assay 2 compared to Assay 1, it still had a Hill coefficient near 2, indicating pre-incubation with spermidine is insufficient to relieve the cooperativity as observed for spermine. Therefore, the enzyme behaves differently in the presence of the asymmetrical polyamine compared to the symmetrical spermine. To determine whether this effect may be explained by a preference for acetylating a specific end of spermidine, we performed LCMS assays to investigate the production of N^1^- or N^8^-acetyl spermidine as described before [29]. Li et al. previously showed the SaSpeG enzyme produced N^1^-acetyl spermine, N^1^-acetyl spermidine, and N^1^-acetyl norspermidine (all aminopropyl ends of the polyamines) and found N^1^-acetyl spermine was a poor substrate for the SaSpeG enzyme [9]. However, it is not known if the enzyme exhibits a preference for the aminobutyl or aminopropyl end of spermidine. Similar to SpeG from *Bacillus thuringenesis* (BtSpeG), we found the SaSpeG enzyme produced both N^1^- and N^8^-acetyl spermidine products in nearly equivalent ratios (Figure 2A,B). Therefore, either end of spermidine can be acetylated by SaSpeG.

We were curious if this ability to acetylate spermidine at either end was specific for Gram-positive bacterial SpeG enzymes or if other Gram-negative SpeG enzymes we had previously characterized also exhibited this behavior. Therefore, we performed the same LCMS assays with the Gram-negative *Escherichia coli* (EcSpeG) and *Vibrio cholerae* (VcSpeG) enzymes. We found the EcSpeG enzyme similarly acetylated N^1^- and N^8^-acetyl spermidine like the Gram-positive BtSpeG and SaSpeG enzymes, but the VcSpeG enzyme showed a strong preference for acetylating the aminopropyl end (N^1^) of spermidine compared to the aminobutyl end (N^8^) (Figure 2C,D). Since the native polyamine in *V. cholerae* is norspermidine, which only has N^1^ aminopropyl groups, the VcSpeG preference for N^1^ spermidine acetylation is consistent with what would be expected biologically in this organism. These results with the EcSpeG enzyme reproduce prior studies of Fukushi et al., who showed either end of spermidine was acetylated [31] and indicate that the preference for aminobutyl or aminopropyl ends of spermidine is not identical across all bacterial SpeG enzymes.

### 3.2. 3D Structure of SaSpeG

To understand more about the SaSpeG enzyme from a structural perspective, we crystallized the protein in the presence and absence of spermine and determined three different structures (Appendix A). We were unable to obtain a structure in complex with spermidine. The ligand-free SaSpeG crystals diffracted to 1.8 Å (Appendix A), and the final model was refined to an *R*_work_ and *R*_free_ of 22% and 24%, respectively (Appendix A) and deposited to the Protein Data Bank (PDB ID 5IX3). In addition, there were two datasets of SaSpeG in complex with spermine obtained at 2.7 Å (PDB ID 8FV0) and 3.0 Å (PDB ID 8FV1). The first dataset contained three SaSpeG protomers in the asymmetric unit with spermine bound in the allosteric site and was refined to an *R*_work_ and *R*_free_ of 22.4% and 27.1%, respectively (Appendix A, PDB ID 8FV0). The second dataset contained four SaSpeG protomers in the asymmetric unit with spermine bound in the allosteric site and was refined to an *R*_work_ and *R*_free_ of 22.5% and 24.4%, respectively (Appendix A, PDB ID 8FV1). The refined crystal structures of all SaSpeG datasets displayed an α/β fold composed of seven central β-strands surrounded by four α-helices (Figure 3A). The β-strands are antiparallel with the exception of parallel strands β4 and β5. The overall topology is composed of β1-α1-α2-β2-β3-β4-α3-β5-α4-β6-β7 (Figure 3B). These interconnected β-sheets formed a V-shape at the interface of the β4 and β5 strands within the protein, which is characteristic of GNATs.

This GNAT fold enables the formation of two distinct sites for substrate binding, where the donor molecule (AcCoA) binds on one face of the V-like splay, and the acceptor molecule binds on the opposite face (Figure 4A). In the case of SaSpeG and other SpeG proteins, a separate allosteric site is also present in an α-helical region located outside of this β-sheet (Figure 4A). In general, binding polyamine to the allosteric site causes a conformational change from a mobile loop to an ordered alpha helix, which is described below for the SaSpeG enzyme and has been observed before for multiple SpeG enzymes from other organisms [28,29]. A comparison of the allosteric site and active site residues of the SaSpeG and VcSpeG protein shows all catalytic and allosteric site residues are conserved, but there is one residue in the active site that differs between these two proteins (Y117 in SaSpeG compared to H122 in VcSpeG) (Figure 4B,C). This residue may be important for orienting the polyamines for N^1^ versus N^8^ acetylation; however, further experiments are required to determine if this hypothesis is supported.

The asymmetric unit of the ligand-free SaSpeG crystal contained a single protomer; however, extensive binding interfaces between symmetry-related proteins and analysis using PISA software [32] suggested the most biologically relevant quaternary structure was a dodecamer (Figure 3C). In addition, SaSpeG crystals in complex with spermine displayed trimeric and tetrameric forms of crystal symmetry, with a predicted dodecameric oligomerization through symmetry mate generation with one spermine molecule bound to each allosteric site (Figure 3D). This finding is consistent with all SpeG structures determined to date, including VcSpeG, EcSpeG, BtSpeG, YpSpeG, and *Coxiella burnetii* CbSpeG. The protein consists of two hexameric rings with D6 symmetry and three types of intermolecular interfaces between adjacent monomers of a single hexamer (IF-1), between monomers of top and bottom hexamers (IF-2), and between diagonal monomers of top and bottom hexamers (IF-3) as described previously for the BtSpeG enzyme [29]. Contacts in the PDB ID 5IX3 structure that are attributed to IF-1 comprise 14 hydrogen bonds, two salt bridges, and a buried interface area of 1073 Å^2^. The salt bridges are formed between residues His^112^ and Glu^142^ and the corresponding symmetry-related interaction (Appendix A). Interactions at IF-2 are composed of 11 hydrogen bonds, two salt-bridge interactions formed between residues Glu^33^ and Arg^52^, and a buried interface of 652 Å^2^. Much weaker interactions were found for IF-3, which was composed of four hydrogen bonds and a buried interface of 414 Å^2^. A full description of the interactions is summarized in Appendix A. Comparison of the apo- and spermine-bound SaSpeG structures revealed a conformational shift within the allosteric loop (residues 18–34) (Figure 5A). The most well-resolved spermine molecule in the allosteric site formed polar interactions with residues Glu^29^, Glu^33^, His^45^, Asp^48^, Glu^49^, and Glu^51^ (Figure 5B). Twelve individual spermine molecules were found in the dodecamer within the allosteric site, similar to previously reported SpeG structures from other bacteria (Figure 5C).

### 3.3. Native PAGE, EMSA, and SEC Analysis of SaSpeG

Our enzyme kinetic results showed the SaSpeG enzyme positive cooperativity was strongly influenced by pre-incubation with polyamines. To determine if this could be due to a possible shift in the oligomeric state, we compared Native PAGE assays, EMSAs, and analytical size-exclusion chromatography (SEC) in the presence and absence of different substrates. First, we incubated the SaSpeG enzyme alone and in the presence of different compounds (spermine, spermidine, cadaverine, putrescine, agmatine, N-acetyl spermine, CoA, and AcCoA) and ran Native PAGE assays. We found a single population of enzymes existed in the presence of individual compounds except for spermine, which revealed the presence of two bands. Therefore, we expanded our analysis to probe whether the different populations of SaSpeG enzyme were influenced by the presence of both substrates (spermine and AcCoA).

We found multiple populations of SaSpeG existed in the presence of spermine (0.25 mM) across different enzyme concentrations (1.25–25 µM), but a single population was obtained once the AcCoA concentration exceeded that of spermine (Figure 6A). Furthermore, when the concentration of the enzyme was held constant at 25 µM, we found multiple populations of SaSpeG were only observed when the concentration of spermine was more than or equivalent to the concentration of AcCoA (Figure 6B). These results indicate the heterogeneity of the SaSpeG enzyme is influenced by both the identity of the polyamine and the concentration of substrates/effectors. We observed similar trends when we performed EMSAs in agarose (Figure 6C). The presence of spermine, but not spermidine or AcCoA alone, showed two populations of SaSpeG. When AcCoA and spermine were in equivalent concentrations, the two populations of SaSpeG were retained, which reproduced the Native PAGE results. The combination of spermidine and AcCoA only showed a single population, indicating only spermine can trigger the formation of heterogeneous populations of SaSpeG.

To determine whether the population shift observed in the native PAGE and EMSAs was due to an alteration in the oligomeric state of the SaSpeG protein, we conducted analytical size-exclusion chromatography in the presence of polyamines or AcCoA and compared the chromatograms (Figure 6D). The ratio of enzyme:polyamine:AcCoA was 1:20:10, and the calculated molecular weight of the SaSpeG monomer is ~19.8 kDa. We found SaSpeG eluted at different volumes (peaks numbered 1–4, corresponding to elution volumes of approximately 14.5, 15.5, 19, and 22 mL, respectively) depending upon the identity of ligands that were present. The peak that was identified at 22 mL is consistent across all samples regardless of ligand and is of very low molecular weight compared to our protein; however, the identity of the low molecular weight species is unknown. Since the peak is consistent across all samples, we focused on the other three peaks for our analysis. SDS-PAGE results showed the samples eluting in pks 1–3 are of the same molecular weight as the SaSpeG monomer (Appendix A). We found the SaSpeG enzyme in the absence of ligands eluted as a single peak (apo; pk1 Figure 6D). This peak shifted toward a later elution volume in the presence of spermine (spm; pk2) and a mixture of two peaks in the presence of spermidine, with the earlier elution peak being most predominant (spd; pk1, pk2) (Figure 6D). Upon addition of AcCoA, two populations of enzymes were present, including one resembling pk1 in the apo chromatogram and a new peak not observed in the presence of polyamines at a later elution volume (AcCoA; pk3; Figure 6D). In the presence of both AcCoA and polyamine (spermine or spermidine), three peaks at different elution volumes were observed and were relatively consistent across both chromatograms (AcCoA+Spm and AcCoA+Spd pk1, 2, 3; Figure 6D). Therefore, the SaSpeG protein exists in multiple oligomeric states in solution depending upon the identity of substrates and/or effectors present.

Prior studies in the laboratory with the same column have shown proteins with molecular weights between 95–100 kDa elute at 11–12 mL, 55 kDa at 14 mL, 45 kDa at 16–17 mL, and 20–30 kDa at 17–18 mL. When we compare the SaSpeG elution volumes (~14.5, 15.5, and 19 mL), the corresponding estimated molecular weight for each peak was approximately 62, 50, and 8 kDa for peaks 1, 2, and 3, respectively. These values were calculated using a calibration curve and the corresponding equation y = −12.162x + 238.68, where x is the retention volume, and y is the calculated molecular weight. We know from previous studies with VcSpeG that binding spermine causes the protein structure to tighten [35]. Therefore, it may be possible that pk2 is a more tightly packed structure than pk1 and appears smaller and elutes later in the chromatogram compared to in the absence of polyamine. Additionally, the peak corresponding to 8 kDa is less than half the total molecular weight of the SaSpeG monomer, indicating molecular weight estimations of the SaSpeG protein appear unreliable by SEC. However, we can glean from these experiments that the size of the SaSpeG protein and quaternary state is altered in the presence of different ligands. Furthermore, these trends are reproducible across different experimental platforms, are concentration dependent, and indicate the oligomeric state of SaSpeG is dynamic.

## 4. Discussion

### 4.1. Oligomeric State of SaSpeG

While our analysis of the SaSpeG protein structure with PISA indicated the most biologically relevant oligomer was a dodecamer, several reports have shown different oligomeric states exist for Gram-negative SpeG proteins. For instance, the oligomeric state of the EcSpeG protein has been reported as a tetramer [31,36], hexamer, and dodecamer [35], and the oligomeric state of the VcSpeG protein was reported as a dimer, tetramer, and dodecamer depending upon protein concentration [35]. Since there have been multiple reports of different oligomeric states of SpeG proteins from various bacteria in the presence and absence of different ligands, we performed size-exclusion chromatography to assess the oligomeric state of the SaSpeG. Contrary to the PISA prediction, we found the SaSpeG protein in the absence of ligands eluted as a single population of trimer/tetramer with analytical and high-prep SEC (~62 kDa and ~70 kDa, respectively; Figure 6D and Appendix A). GNATs exist in a variety of oligomeric states, ranging from monomers, dimers, tetramers, and dodecamers; however, to our knowledge, a GNAT trimer has never been reported. Based on the dodecameric assembly predicted by PISA and other GNAT studies, we think it is unlikely a trimer would be the oligomeric state of the SaSpeG enzyme. We hypothesize the estimated molecular weight may be inaccurate due to interactions of the protein with the column based on the structural characteristics of the SaSpeG protein described below.

Since SEC is commonly used to separate proteins by size, we often do not consider that the protein could interact with the column and alter retention times. Superdex resin is composed of porous agarose beads that are cross-linked to dextran and are commonly used to estimate the molecular weights of proteins. Early studies on Superdex resins showed that ionic and hydrophobic interactions were possible and affected small molecule retention times [37,38]. While ionic protein interactions can be minimized by increasing NaCl concentrations, hydrophobic interactions can increase in the presence of NaCl. Because we observed longer retention times with the SaSpeG enzyme compared to previous reports with other SpeG proteins, we considered the electrostatic or hydrophobic properties of the SaSpeG enzyme may be different. Therefore, we generated electrostatic surfaces of SaSpeG, BtSpeG, VcSpeG, EcSpeG, and YpSpeG crystal structures in the presence and absence of spermine (Figure 7).

When the apo and liganded structures of SaSpeG were compared, we found the overall electrostatic surface of the protein changed even though the structures were determined from crystals that were grown at the same pH. The internal acidic residues radiate from the center of the dodecamer in the absence of ligands but become more centralized in the spermine-bound structures. Upon binding spermine in the allosteric site, basic residues become surface exposed in an extensive circular pattern compared to patches of basic residues on other SpeG dodecamers. We also examined the placement of small hydrophobic and aromatic residues within the dodecameric structures of different SpeGs. While all of these proteins had large numbers of hydrophobic and aromatic residues within their sequences (41–45% across SpeGs; Table 2), the placement of aromatic residues differed across SpeGs. Interestingly, SaSpeG showed a striking set of radiating hydrophobic and aromatic residues at the interfaces of monomers of the hexamer in the spermine-bound structure. Thus, while the total number of these residues is quite similar across SpeGs, their placement appears distinctive for the SaSpeG enzyme and may contribute to its in vitro behavior compared to other SpeGs previously characterized. Clearly, further study of the dynamic oligomerization of SaSpeG with other experimental techniques is required.

### 4.2. Differential Effects of Spermine and Spermidine on SaSpeG Cooperativity and Heterogeneity

Several complementary experiments herein revealed that spermine has the capacity to alter both the kinetic and structural behavior of the SaSpeG enzyme. Interestingly, this behavior is polyamine dependent, where incubation with spermine produces protein heterogeneity and Michaelis–Menten behavior. On the other hand, SaSpeG shows positive cooperativity toward spermidine and does not produce heterogeneous populations of enzymes in Native PAGE or EMSAs. Structural studies have revealed the presence of a polyamine allosteric site between protomers of SpeG hexamers. Therefore, positive cooperativity could reflect either polyamine binding to the allosteric site, assembly of oligomers, or both. Kinetic assays show the catalytically functional enzyme resembles nearly identical apparent affinities toward both spermine and spermidine after pre-incubation with the corresponding polyamine. This could imply the retained positive cooperativity after pre-incubation may be due to a lower binding affinity of the allosteric site for spermidine compared to spermine due to the shorter length and inability to span binding to two protomers like spermine. Furthermore, it may explain why we were unable to obtain a crystal structure with spermidine in the allosteric site. However, further biophysical studies are required to determine if this data support the binding affinity hypothesis.

We found SaSpeG homogeneity is affected by concentrations of AcCoA and the identity of the polyamine. Higher concentrations of spermine than AcCoA promote multiple and higher-ordered states, but when AcCoA concentrations are higher than spermine, the protein appears more homogeneous and lower-ordered. However, we were unable to shift the entire population of the enzyme toward a homogeneous higher-ordered oligomer even under increased concentrations of spermine. It is currently unclear why this is the case unless the protein itself is heterogeneously post-translationally modified and affects oligomeric assembly. Moreover, it is unclear whether multiple oligomeric states are catalytically operational or if there is an oscillation between states depending upon polyamine concentrations or the energy state (AcCoA) within the cell. To our knowledge, the only other bacterial GNAT that has been reported to exhibit positive cooperativity and affect oligomerization is the YfiQ/PatZ lysine acetyltransferase protein from *Salmonella enterica* [39]. This positive cooperativity was observed toward AcCoA, which affected monomer to tetramer YfiQ/PatZ oligomerization. Parallels between these proteins (Yfiq/PatZ and SpeG) may indicate an underrecognized role of GNAT substrates on protein oligomerization mechanisms.

### 4.3. Comparison of SaSpeG Kinetics

When we compared the kinetic results of the SaSpeG enzyme in this study with those of Li et al. [9], all assays showed the enzyme had a higher apparent affinity, k_cat_, and catalytic efficiency toward spermine compared to spermidine (Table 1). However, there were discrepancies in the magnitude of different kinetic parameters. For example, we found the apparent affinity of SaSpeG for spermine was approximately 4-fold lower with Assay 1 and 4-fold higher with Assay 2 compared to Li et al. Similarly, the apparent affinity for spermidine was 3-fold lower with Assay 1 and 21-fold higher with Assay 2 compared to Li et al. The turnover number of the enzyme toward both spermine and spermidine was comparable between Assay 1 and Li et al. but increased by one order of magnitude in Assay 2. This resulted in the catalytic efficiencies toward spermine and spermidine being 2–3 orders of magnitude higher for Assay 2 compared to previous results.

While we do not know the exact cause for discrepancies in the two studies, we know, based on the results presented herein, that the SaSpeG enzyme is sensitive to different reaction conditions. There are some dissimilarities in the protein preparations and assays that may contribute to this. For example, the protein used by Li et al. was dialyzed into a buffer containing 2 mM CaCl_2_ after purification and assayed in the presence of 1 mM EDTA in Tris HCl pH 7.5. Additionally, to our knowledge, the polyhistidine tag was not cleaved for the Li et al. studies, whereas it was cleaved for our assays. Furthermore, the *speG* gene was cloned and expressed in different vectors with different antibiotic resistances (kanamycin versus spectinomycin) and the identities of the residues that comprise the N-terminal polyhistidine tag, spacer residues, and TEV cleavage sites are also different (pMCSG21: MHHHHHHSSGVDLGTENLYFQ/SN (this study) compared to pET28-TEV: MGSSHHHHHHSSGENLYFQ/GH (Li et al. study)). We previously showed the VcSpeG enzyme activity was affected by the polyhistidine tag presence [40], but it is not known whether the same applies to SaSpeG. In this study, the polyhistidine tag on the SaSpeG enzyme was removed for all kinetics assays and structure determination experiments. Further studies are currently underway in our laboratory to discern whether these parameters affect recombinant SpeG enzyme biochemical and biophysical characteristics.

### 4.4. Dynamic Role of SaSpeG

SpeG acetylates polyamines to regulate their cellular concentrations, but its role may be beyond catalytic and could additionally function in a moonlighting capacity, such as forming protein–protein interactions with other cellular partners to regulate additional processes. Indeed, its ability to oscillate between different oligomeric states based on the identity of specific polyamines and concentrations of substrates renders the concept of an alternative regulatory role attractive. Support for additional roles of *speG* has been identified in *Salmonella enterica* serovar Typhimurium, where *speG* not only affects cellular polyamine concentrations but also regulates a variety of genes important for flagellar biosynthesis, glucarate metabolism, and histidine biosynthesis, among others; it is also required for *Salmonella* replication in epithelial cells [41]. Additionally, *speG* expression is important for clindamycin antibiotic resistance in *B. licheniformis* and *B. paralicheniformis,* but it is unclear whether a catalytically competent enzyme is required [42]. The new structural and additional kinetic insight presented in this study for SaSpeG provides a platform for future exploration into its roles in *S. aureus* as well as other antimicrobial-resistant bacterial pathogens.

## Figures and Tables

**Figure 1 cells-12-01829-f001:**
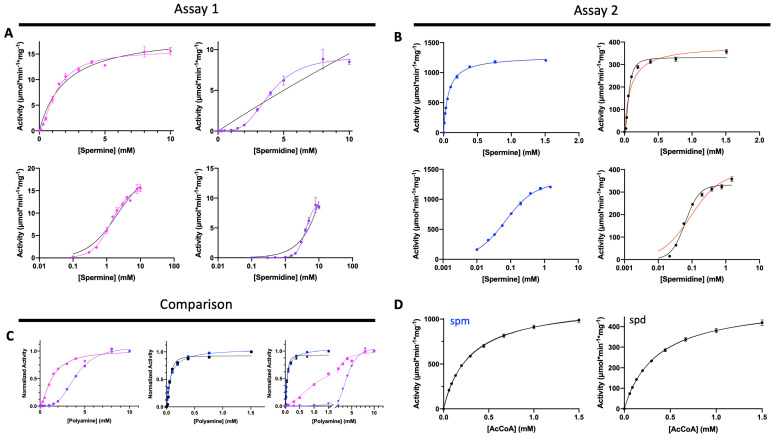
Kinetic characterization of SaSpeG using two different assays. The difference between Assay 1 and 2 was that the SaSpeG enzyme was not pre-incubated with polyamine prior to Assay 1 compared to pre-incubation in Assay 2. See Materials and Methods for additional assay details. (**A**) Assay 1 results of polyamine substrate saturation curves. Data were fitted with both Michaelis–Menten (MM) and an allosteric sigmoidal equation in Prism 8.0. The MM equation fitting is indicated with a black line, whereas fitting with allosteric sigmoidal is in pink for spermine and purple for spermidine. Log plots (second row) are shown to clarify the fitting between the two equations. (**B**) Assay 2 results of polyamine substrate saturation curves. Similar to panel A, data were fitted to two equations (MM and allosteric sigmoidal), and log plots are shown. The MM equation fitting is indicated with a red line, whereas fitting with the allosteric sigmoidal equation is shown in blue for spermine and black for spermidine. (**C**) Comparison of Assay 1 and Assay 2 results after normalization for both polyamines. Curves are colored as mentioned in panels A and B. (**D**) Substrate saturation curves for AcCoA in the presence of spermine (spm) and spermidine (spd) as described in Materials and Methods.

**Figure 2 cells-12-01829-f002:**
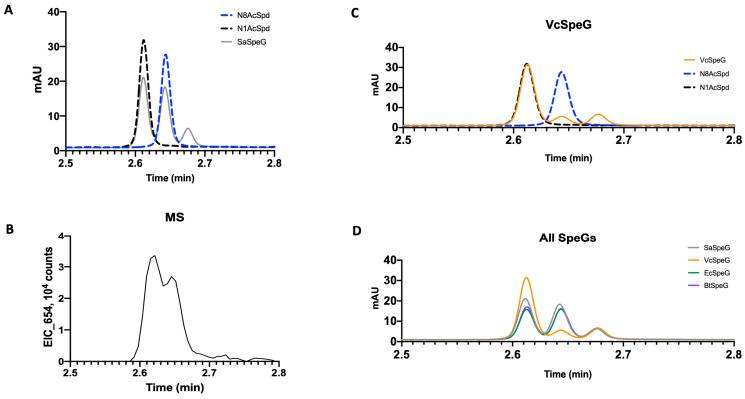
Products of enzymatic spermidine acetylation by SpeG enzymes. (**A**) UV chromatograms (293 nm) of the acetylated products of a 5 min reaction (solid gray line) of SaSpeG overlaid with the injections of 200 µM standards of N^1^-acetylspermidine (black dashed line) and N^8^-acetylspermidine (blue dashed line). (**B**) Extracted ion chromatogram of the reaction products corresponding to a proton adduct of the dansylated acetylspermidine (*m*/*z* 654). (**C**) UV chromatograms (293 nm) of the acetylated products for VcSpeG (solid orange line) compared to standards as in panel A. (**D**) UV chromatograms (293 nm) of the acetylated products for all tested SpeGs. SaSpeG (gray), VcSpeG (orange), EcSpeG (green), and BtSpeG (purple). Data for BtSpeG is the same as previously reported in [29]. Ratios of N8-acetylspermidine to N1-acetylspermidine products were SaSpeG (1:1.13), VcSpeG (1:6.23), EcSpeG (1:0.97) and BtSpeG (1:1.02).

**Figure 3 cells-12-01829-f003:**
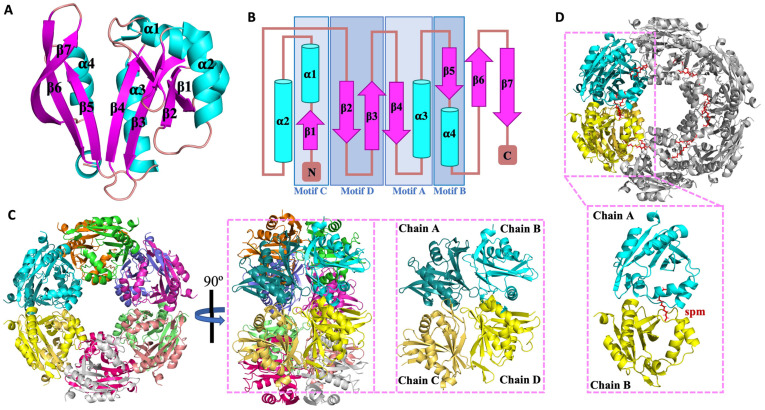
Structure of SaSpeG. (**A**) SaSpeG protomer displays an α/β GNAT fold. (**B**) Topological representation of SaSpeG protomer consists of 4 α-helices and 7-β strands with GNAT motifs A–D highlighted. (**C**) SaSpeG dodecamer is a functional unit of the enzyme formed by a dimer of hexamers. Two GNAT dimers, chain A and chain B, form one GNAT dimer, and chain C and chain D form the other. Interface-1 interactions lie within GNAT dimer, chain A and chain B or chain C and chain D, Interface-2 interactions between chain B and chain D or chain A and chain C, Interface-3 interactions are present between chain A and chain D or chain B and chain C. (**D**) Two SaSpeG protomers of the same hexamer with bound spermine molecule in the allosteric site.

**Figure 4 cells-12-01829-f004:**
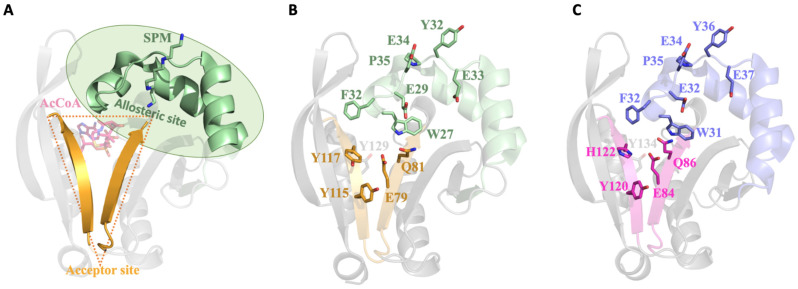
Comparison of SaSpeG and VcSpeG acceptor and allosteric sites within a single monomer. (**A**) SaSpeG protein (PDB ID 8fv1 chain B) with AcCoA modeled from the VcSpeG PDB ID 4r57 structure. The donor site contains AcCoA in magenta, the allosteric site in complex with spermine is shown in green, and the V-splay of the acceptor site is highlighted in gold. (**B**) SaSpeG PDB ID 8fv1 chain B structure with allosteric and acceptor site residues highlighted as described in panel A. The conserved catalytic tyrosine residue Y129 is shown in grey. (**C**) VcSpeG PDB ID 4r57 chain A structure with allosteric site residues in purple and active site residues in magenta, and critical catalytic Y134 residue is in grey.

**Figure 5 cells-12-01829-f005:**
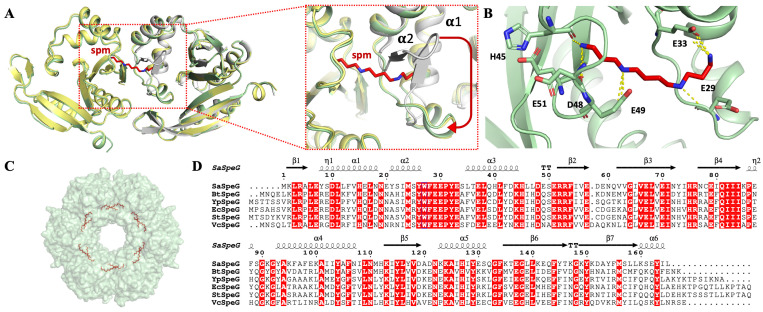
SaSpeG structure in complex with spermine and representative Gram-positive and Gram-negative SpeG protein sequence alignment. (**A**) Structural alignment of the SaSpeG apo form (gray, PDB ID 5IX3) and SaSpeG in complex with spermine (spm) (yellow and light green PDB ID 8FV0 and 8FV1, respectively) demonstrating the flexibility of the loop between helices α1 and α2 to accommodate the binding of spermine in the allosteric site. Loop movement is indicated with a red arrow. (**B**) Residues E29, E33, H45, D48, E49, E51 form interactions with spermine. (**C**) Surface of the dodecamer with twelve spermine molecules positioned within the allosteric sites of each monomer within the dodecamer. (**D**) Multiple sequence alignment using MultAlin [33] and then visualized using ESPript 3.0 [34]. Protein sequences include *Staphylococcus aureus USA300* (SaSpeG), *Bacillus thuringenisis* (BtSpeG), *Yersinia pestis* (YpSpeG), *Escherichia coli* (EcSpeG), *Salmonella typhimurium* SL1344 (StSpeG), and *Vibrio cholerae* (VcSpeG) from UniProt IDs: A0A0H2XGJ0, A0A437SL45, A0A5P8YIA2, P0A951, A0A719HIP0, and Q9KL03, respectively. The secondary structure of the SaSpeG protein crystal structure (PDB ID 8FV0) is shown above the alignment. Conserved amino acids are highlighted in red.

**Figure 6 cells-12-01829-f006:**
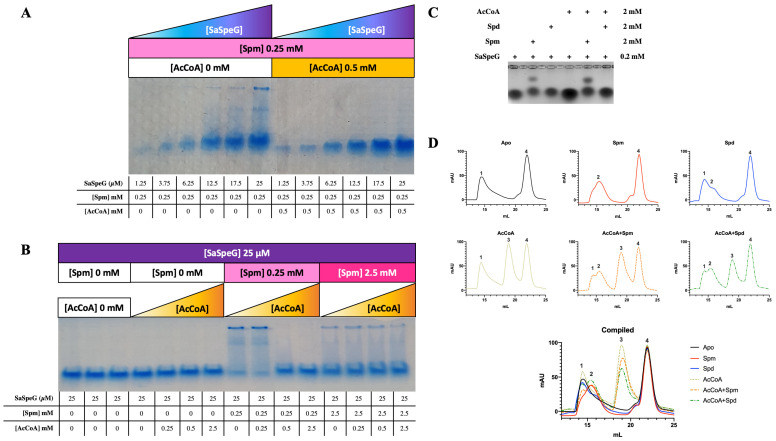
Evaluation of SaSpeG oligomerization using Native PAGE, EMSA, and SEC. (**A**,**B**) Native PAGE of SaSpeG at different concentrations of protein and in presence or absence of different concentrations of spermine (spm) and/or AcCoA. Protein was purified from cells not grown in auto-induction media. (**C**) EMSA of SaSpeG in presence and absence of spm, spermidine (spd), and/or AcCoA. Protein was purified from cells grown in auto-induction media. (**D**) Size-exclusion chromatograms of the SaSpeG protein in presence and absence of spermine (spm) produced four elution peakes labeled 1–4, respectively.

**Figure 7 cells-12-01829-f007:**
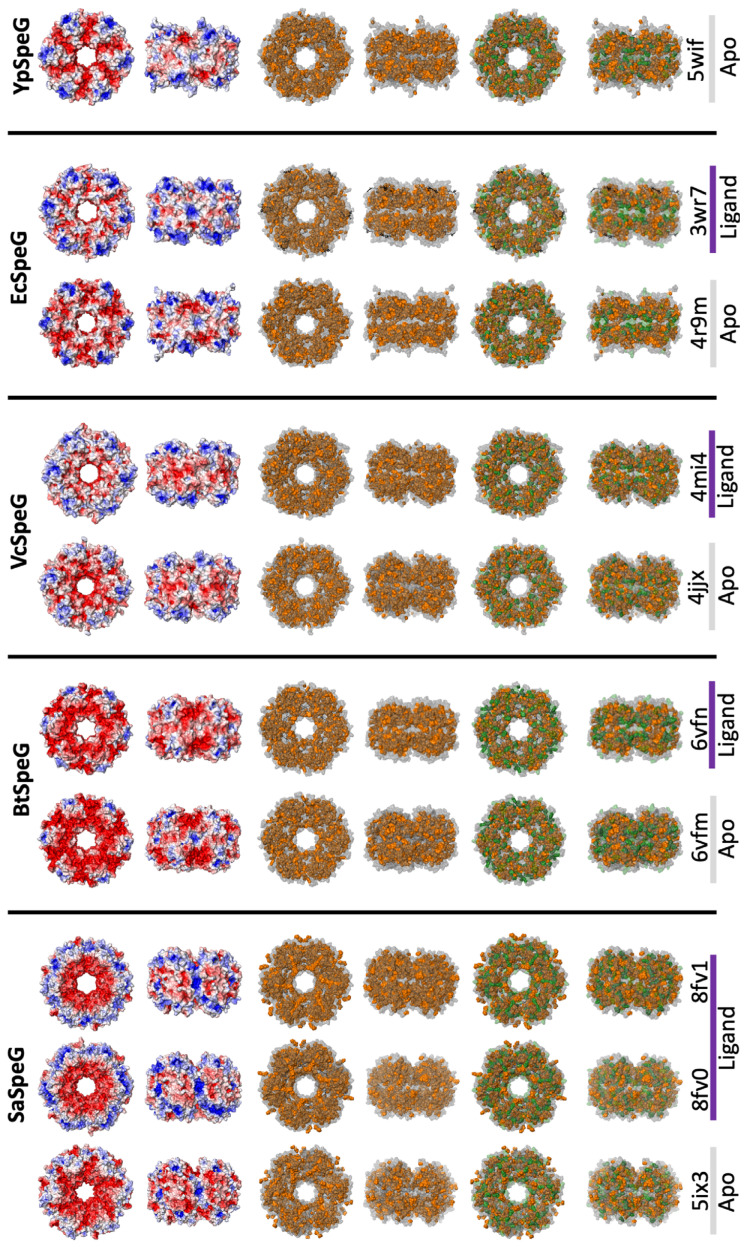
Electrostatic and hydrophobic surfaces of Gram-positive and Gram-negative bacterial SpeG crystal structures. Dodecamer electrostatic surfaces (**top** two rows) are shown in red (negative), blue (positive), and white (neutral). Distribution of hydrophobic and aromatic residues (alanine, valine, isoleucine, leucine, phenylalanine, tyrosine, and tryptophan) on dodecamers are shown in orange (**middle** two rows). Distribution of aromatic residues (phenylalanine, tryptophan, tyrosine) in green compared to small hydrophobic residues (alanine, valine, isoleucine, leucine) in orange (**bottom** two rows). The full surface of the protein is shown in gray. Top and side (rotated 90º along y-axis) views of each dodecamer are shown for each representation. PDB IDs are listed beneath columns of structures; apo structures are shown with gray bars, and liganded structures with purple bars. All liganded structures were in complex with spermine, with the exception of 3wr7, which was in complex with both spermine and CoA.

**Table 1 cells-12-01829-t001:** SaSpeG kinetic parameters toward spermine (spm), spermidine (spd), and AcCoA. Results of two different assays described in Materials and Methods are shown. All substrate saturation curves were fitted with the Michaelis–Menten equation and the allosteric sigmoidal equation in Prism 8.0. Only kinetic parameters for the best fitting equation are shown (e.g., K_m_ for Michaelis–Menten and S_0.5_ for allosteric sigmoidal). Red text indicates K_m_, whereas black text indicates S_0.5_, which describes the concentration of substrate being tested to reach half the maximal velocity. The Hill number (n) is only shown for allosteric sigmoidal fittings. Data previously reported by Li et al. [9].

		K_m_ or S_0.5_ (mM)			k_cat_ (s^−1^)	Catalytic Efficiency (M^−1^s^−1^)	n
Assay 1							
	Spm	1.3	±	0.1	5.3	4.1 × 10^3^	1.6 ± 0.2
	Spd	4.0	±	0.1	3.0	7.5 × 10^2^	3.5 ± 0.2
Assay 2							
	Spm	0.070	±	0.001	420	6.0 × 10^6^	
	Spd	0.063	±	0.005	110	1.7 × 10^6^	2.1 ± 0.3
	AcCoA (spm)	0.29	±	0.01	390	1.3 × 10^6^	
	AcCoA (spd)	0.35	±	0.01	170	4.9 × 10^5^	
Li et al.							
	Spm	0.295	±	0.005	3.52	1.2 × 10^4^	
	Spd	1.33	±	0.12	2.48	1.8 × 10^3^	

**Table 2 cells-12-01829-t002:** Amino acid composition of SpeG proteins. Comparison of the number of small hydrophobic and aromatic residues present and percentage of the total number of amino acids in each SpeG monomer. One-letter amino acid abbreviations are shown for small hydrophobic and aromatic amino acids (AAs). The calculated isoelectric point of each protein is also indicated. Data were obtained using the Expasy Server ProtParam tool (https://web.expasy.org/protparam/, accessed on 1 March 2022).

	pI	#AAs	#A, V, I, L	#F, W, Y	% Small Hydrophobic AAs	% Aromatic AAs	% Small Hydrophobic and Aromatic AAs
SaSpeG	5.28	165	8, 7, 16, 18	11, 1, 14	30	16	45
BtSpeG	5	171	9, 14, 10, 14	11, 1, 12	27	14	42
VcSpeG	5.58	173	9, 9, 15, 19	10, 1, 10	30	12	42
EcSpeG	6.2	186	13, 12, 11, 19	9, 1, 11	30	11	41
YpSpeG	5.99	181	11, 8, 18, 16	10, 1, 11	29	12	41

## Data Availability

The structures and experimental data have been deposited to the Protein Data Bank and issued the codes 5IX3, 8FV0, and 8FV1.

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
