# Peer review of "Structural and Kinetic Characterization of the SpeG Spermidine/Spermine N-acetyltransferase from Methicillin-Resistant Staphylococcus aureus USA300"

_cells, 2023, doi:10.3390/cells12141829_

Round 1

Reviewer 1 Report

In their study, Tsimbalyuk and colleagues introduce the crystal structures of SpeG polyamine acetyltransferase derived from MRSA, both in the presence and absence of the ligand and elucidate their kinetic properties. While this structure is similar to previously identified structures of polyamine acetyltransferases from various bacteria, it manifests a significant divergence in the amino acid sequence. This distinction justifies a comprehensive analysis, particularly as it proposes a potential target in the fight against multi-drug resistant (MDR) bacteria. Given the importance and implications of these findings, I advocate for the publication of this paper.

Some minor comments for consideration are as follows:

1. The paper is all about the enzyme, but the active site is not described. I suggest the authors introduce a paragraph in the results section elucidating this, along with a figure depicting the active site cleft, compared to the most closely related SpeG.

2.     The activity of SpeG seems to significantly differ based on whether SpeG is preincubated with the polyamine prior to the assay or not. This difference seems logical, as polyamine acetylation is only required when polyamines are present. However, the issue arises because polyamine is both a substrate and an allosteric modifier. Consequently, the minimal activity in Assay 1 might be ascribed to the substrate (polyamine) that also attaches to the allosteric site. Therefore, SpeG might be inactive without polyamines in the allosteric sites. It would be insightful to compare the structures of SpeG's active sites with and without polyamine, as the authors have solved but not compared these structures in that regard. This comparison could help determine if binding in the allosteric site may influence the active site.

3.     The authors examine the oligomeric state of the enzyme using size exclusion chromatography, offering elution volumes for the protein in its presumed different oligomeric states. They also provide reference sizes based on their analysis of various other proteins on the same column (line 527). It would be beneficial if they strive to derive a calibration curve from this data or construct one, offering estimated molecular weights instead of elution volumes.

4.     A discussion concerning the N-terminal of the SpeG, which may affect its catalytic activity, is included. It would be interesting to see if this N-terminal is proximate to the active site, allosteric site, or if it can influence the oligomeric state of the enzyme (devised from the structure).

Additional recommendations:

a) Figure 6 could be larger for better visibility.

b) A reference is required in line 42, Murray, 2019 from Lancet?

c) The Phaser reference (line 140) is absent from the reference list.

d) The formatting of the references in lines 168 and 190 requires correction.

Reviewer 2 Report

There is not sufficient back ground for the research, previous investigations are in collision and there are not enough evidence about this variation and their role in AMR. In general the basic idea of research is not listed. Methods are very confusing and extremely hard to read with large quantity of unexplained abbreviations. If the reader is not a lab tech it is very hard to understand it. Results are not following the methods in 100% it looks like some new ideas are on the authors minds in this section.  Discussion are the biggest problem, whole sections of it have no comparations with other authors, like line 609 and 610, and the rest is mostly resting on just one - Li et al. 2019. Also there are several provisional comments on the results with no back up. Conclusion is not mandatory and it can be in the discussion but here is literally not existing.

English is fine.

Reviewer 3 Report

The authors have characterized the SpeG enzyme of Staphylococcus aureus pathogenic strain USA300. SpeG is a spermine/spermidine N-acetyltransferase, and the SpeG of S. aureus USA300 has been linked to its rapid spread and virulence. After the human spermine/spermidine N-acetyltransferase, the S. aureus USA300 SpeG is therefore the most consequential spermine/spermidine N-acetyltransferase for human health. A structure and kinetic characterization of the S. aureus USA300 strain SpeG protein is an important contribution to understanding not just the success of USA300 as a pathogen, but also more fundamentally how spermine N-acetylation has evolved differently in different pathogens.

The SpeG structure presented by the authors includes a comparison of the enzyme with and without spermine. One of the intriguiging aspects of this work is that the authors have shown a dynamic oligomeric state of SpeG according to the concentration of the substrate spermine. Like other SpeG enzymes USA300 SpeG has an allosteric site bound by spermine but the USA300 SpeG is more structurally complex in its higher order structure and is 100 times more active if pre-incubated with spermine, suggesting structural response to spermine binding resulting in increased catalytic efficiency. The current work reveals a previously unsuspected sophistication to SpeG behavior and opens the door to more detailed dissection of this important enzyme. The work has been well executed and clear lines of thinking are evident in the experimental approach. In addition, the manuscript is well written, clear and easy to read.

I have only few minor comments.

1.       Line 235. The authors used 1 ng of enzyme (monomer 19.8 kDa) for each assay reaction (in 100 microlitre volume). If my calculation is correct, this is 500 picomolar enzyme – could the authors please check that this is the correct amount of enzyme?

2.       Line 63. Spermine is a tetraamine, not a triamine.

3.       Line 33. “aminopropyl” rather than “amiopropyl”.

4.       Line 396. “…but the VcSpeG enzyme showed a strong preference for acetylating the aminopropyl end (N1) of spermidine..”. Perhaps the authors could add that this is consistent with Vibrio cholerae having norspermidine, which has only N1-aminopropyl groups, as its native triamine.

5.       Line 527. “Prior studies [in] the laboratory”.

6.       Line 609. [complementary] rather than [complimentary], although we all like compliments.

7.       The authors appear to have forgotten to format the references.
